

# On the embryonic cell division beyond the contractile ring mechanism: experimental and computational investigation of effects of vitelline confinement, temperature and egg size

Evgeny Gladilin[1,2], Roland Eils[1,2] and Leonid Peshkin[3]

[1] Theoretical Bioinformatics, German Cancer Research Center, Heidelberg, Germany
[2] BioQuant and IPMB, University Heidelberg, Heidelberg, Germany
[3] Systems Biology, Harvad Medical School, Boston, MA, USA

Corresponding authors
Evgeny Gladilin, e.gladilin@dkfz.de
Leonid Peshkin,
pesha@hms.harvard.edu

## ABSTRACT

Embryonic cell division is a mechanical process which is predominantly driven by contraction of the cleavage furrow and response of the remaining cellular matter. While most previous studies focused on contractile ring mechanisms of cytokinesis, effects of environmental factors such as pericellular vitelline membrane and temperature on the mechanics of dividing cells were rarely studied. Here, we apply a model-based analysis to the time-lapse imaging data of two species (*Saccoglossus kowalevskii* and *Xenopus laevis*) with relatively large eggs, with the goal of revealing the effects of temperature and vitelline envelope on the mechanics of the first embryonic cell division. We constructed a numerical model of cytokinesis to estimate the effects of vitelline confinement on cellular deformation and to predict deformation of cellular contours. We used the deviations of our computational predictions from experimentally observed cell elongation to adjust variable parameters of the contractile ring model and to quantify the contribution of other factors (constitutive cell properties, spindle polarization) that may influence the mechanics and shape of dividing cells. We find that temperature affects the size and rate of dilatation of the vitelline membrane surrounding fertilized eggs and show that in native (not artificially devitellinized) egg cells the effects of temperature and vitelline envelope on mechanics of cell division are tightly interlinked. In particular, our results support the view that vitelline membrane fulfills an important role of micromechanical environment around the early embryo the absence or improper function of which under moderately elevated temperature impairs normal development. Furthermore, our findings suggest the existence of scale-dependent mechanisms that contribute to cytokinesis in species with different egg size, and challenge the view of mechanics of embryonic cell division as a scale-independent phenomenon.

## INTRODUCTION

Cell division is an indispensable part of the natural life cycle of all eukaryotic organisms. Despite an apparent geometrical simplicity, embryonic cell division is a complex, multi-step process driven and supported by several contributing mechanisms, including

- contraction of the cleavage furrow (*Swann & Mitchison, 1958*),
- polarization of mitotic spindle (*Dan, 1958*),
- astral relaxation of membrane tension (*Wolpert, 1960*; *Wolpert, 1963*),
- expansion of cell surface (*Nachtwey, 1965*),
- lipid vesicles trafficking (*Schiel & Prekeris, 2010*; *Neto, Collins & Gould, 2011*).

Based on experimental observations of dividing embryonic egg cells (*Hiramoto, 1958*; *Schroeder, 1968*; *Rappaport, 1971*), attempts were undertaken to quantitatively characterize the contribution of several of intrinsic mechanisms of cytokinesis. Plausibility models of spindle polarization and astral membrane relaxation have been presented in *White & Borisy (1983)*, *Akiyama, Tero & Kobayashi (2010)*. Continuum mechanics models (*Greenspan, 1977*; *Pujara & Lardner, 1979*; *Akkas, 1981*) showed that experimentally observed cell shape changes during cytokinesis can be explained by a passive material response of cellular matter to the impact of contractile forces in the equatorial furrow region. *He & Dembo (1997)* combined the formalism of astral signal triggering on the embryonic surface with computational simulation of cellular elasticity. Despite a general agreement with experimental data, mathematical models of contractile ring mechanism (*Akkas, 1981*; *He & Dembo, 1997*) fail to explain the polar elongation of sea urchin (*Clypeaster japonicus, Cj*) (*Hiramoto, 1958*), which is significantly larger than predicted.

In addition to intrinsic cytokinetic mechanisms, physical cell environment has a distinctive impact on embryonic cell division. Two obvious physical factors that can principally affect cytokinesis are (i) temperature and (ii) the natural mechanical cell confinement as given by the periembryonic vitelline membrane. Surprisingly, effects of the physical environment on cytokinesis have been rarely investigated. The vitelline membrane has usually been considered an obstacle to be removed prior to investigation of egg cleavage (*Hiramoto, 1958*; *Koyama et al., 2012*). The impact of vitelline membrane as a mechanical confinement on embryonic cell division has never been investigated. Previous works have shown the effects of temperature on mechanical properties of fiber-rich biological gels (*Tempel, Isenberg & Sackmann, 1996*; *Xu, Wirtz & Pollard, 1998*; *Semmrich et al., 2007*), suspended (*Chan et al., 2014*) and embryonic cells (*Marsland & Landau, 1954*; *Mitchison & Swann, 1954*) as well as timing of cell division (*Begasse et al., 2015*). However, the effects of temperature on division of vitelline-confined embryonic cells have not been investigated.

In this study, we build on our earlier success in computational image processing and biomechanical modeling of cellular structures (*Gladilin et al., 2007*; *Gladilin et al., 2008*) and apply similar ideas to develop a novel two-pronged approach to quantitative analysis of the effects of vitelline confinement, temperature and egg size on mechanics of the first embryonic division. We perform an image- and model-based investigation of cytokinesis

of two model species with large eggs, i.e., acorn worm (*Saccoglossus kowalevskii*, *Sk*) and African clawed frog (*Xenopus laevis*, *Xl*), that differ in the type of their mechanical vitelline confinement and are experimentally exposed to different temperatures. Furthermore, we apply our model to describe the polar elongation of small egg cells such as *Cj*.

## METHODS

### Image acquisition

*Sk* embryos were obtained by standard *in vitro* fertilization protocol (*Lowe et al., 2004*) using eggs from a single female and sperm form a single male. The temperature was controlled by placing the glass Petri dish on a metal stage with a constant heat exchange by circulating water from a controlled temperature water bath. Over the duration of the experiments, the sample temperature did not deviate from described value by more than 0.2 °C as controlled by a 4,238 Traceable thermocouple thermometer certified to a resolution of 0.1 °C and accuracy of 0.3 °C. The time-lapse imaging at 4 frame/second was done with a regular Nikon camera mounted on a dissection microscope with 10× magnification lens (Fig. 1). The embryos were kept on to ensure normal development. Effects of temperature on cytokinesis are investigated on the basis of time-lapse image series of *Sk* embryonic cells that exhibit visible polar elongation during the cleavage. To ensure data consistency, only cells that undergo symmetric cleavage in the plane orthogonal to the field of view are preselected for subsequent image analysis. Out-of-plane and non-symmetrically dividing cells are excluded from analysis.

### Image processing

3D (2D + time) stacks of each experimental time series of images are denoised and semi-automatically segmented with the help of Amira v4.1 (Mercury Computer Systems, Arlington, VA, USA), see Fig. 2A. Subsequently, spatial–temporal isosurfaces (Fig. 2B) and contours (Fig. 2C) of dividing cells are generated for all previously segmented cells using Amira's surface and contour generating routines. For all cells and all time steps $s = 1..N$, coordinates of mass center points and the lengths of the shortest (equatorial furrow) $F(s)$ and longest (polar) $L(s)$ embryonic axes are calculated using a C computer program developed in-house (Fig. 2D). Time-series of $F(s)$ and $L(s)$ are smoothed using the 5-point masked median filter for subsequent computation of time-derivatives $(F'(s), L'(s))$ and detection of time steps of the image sequence $s \in [s_s, s_e]$ corresponding to the first embryonic cell division. For this purpose, the sum of absolute derivatives $(SAD = |F'(s)| + |L'(s)|)$ is calculated which serves as an indicator of local curve steepness (Fig. 3). Start $(s_s)$ and end $(s_e)$ time points of the first embryonic cleavage are determined automatically as arguments of local minima left and right from the absolute maximum of $SAD$ and subsequently validated by visual inspection. For an invariant description of cytokinesis, a dimensionless time $t = (s - s_s)/(s_e - s_s) \in [0, 1]$ is introduced.

### Geometrical modeling

For simulating cellular deformations during the cleavage, a spherically-shaped 3D triangulated surface model is generated. The closed surface is filled up with an unstructured

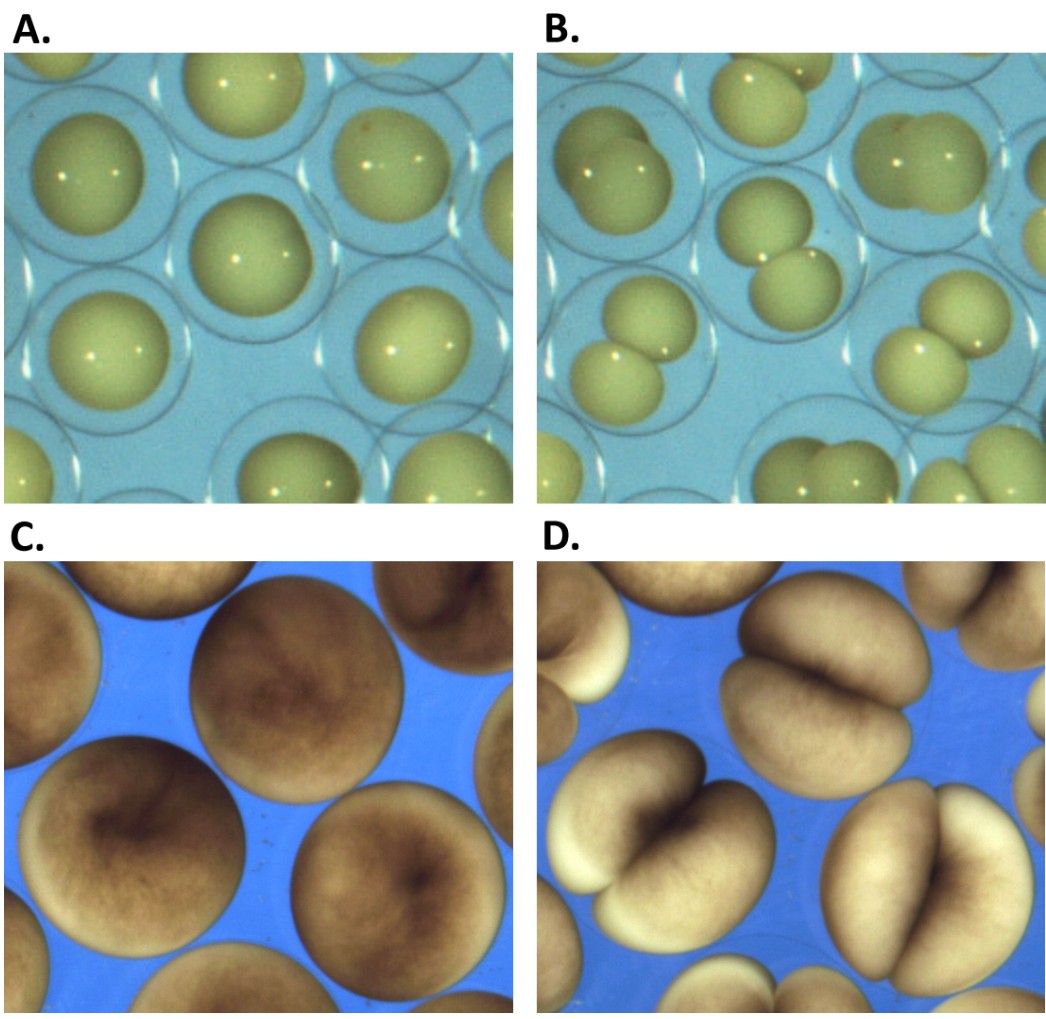

**Figure 1 Embryonic cell division.** (A, C) show the begining and (B, D) the end of the first embryonic division of *Sk* (A, B) and *Xl* (C, D) cells. While vitelline unconstrained *Sk* cells demonstrate free extension of their polar length during the cleavage, tight mechanical confinement restrains *Xl* cells to deform within the vitelline sphere.

tetrahedral grid using Amira's TetraGen tool. For every new step of the multi-step simulation procedure, the surface generation and tetrahedral grid generation is repeated anew.

Dimensions of all essential geometrical parameters required for simulation of cell division (such as cell cross-section, furrow width, spindle length, etc.) are converted from the physical scale (i.e., in μm) into the dimensionless scale of the virtual cell model according to the relative proportions, see example in Table 1.

## Physical modeling

Following the assumption of the contractile ring theory, we initially model the first embryonic cell division as a deformation of a three-dimensional elastic ball successively constricted in its equatorial plane by the contracting cleavage furrow. Starting from this

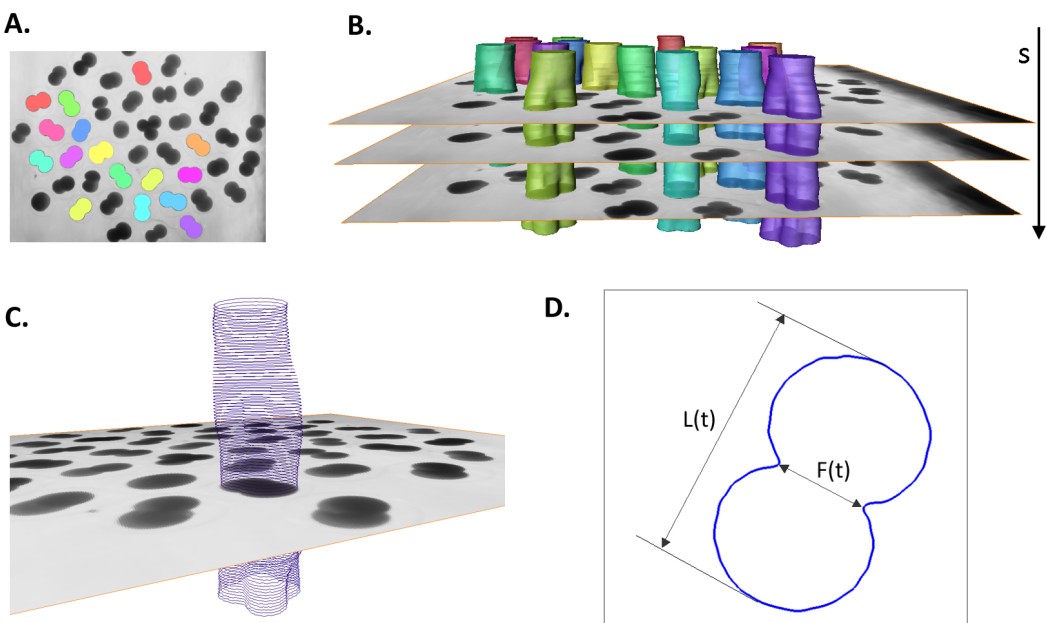

**Figure 2 Image processing and contour extraction.** (A) shows segmentation of *Sk* embryonic cells (colored areas and isosurfaces) in a single 2D microscopic image and (B) in entire 3D (2D image + time) stack. (C) Cellular contours are extracted from the boundary of segmented cells. (D) To quantify the cellular shape at every time step ($s$) of the image sequence, the lengths of the furrow $F$ and the embryonic polar axis $L$ are calculated.

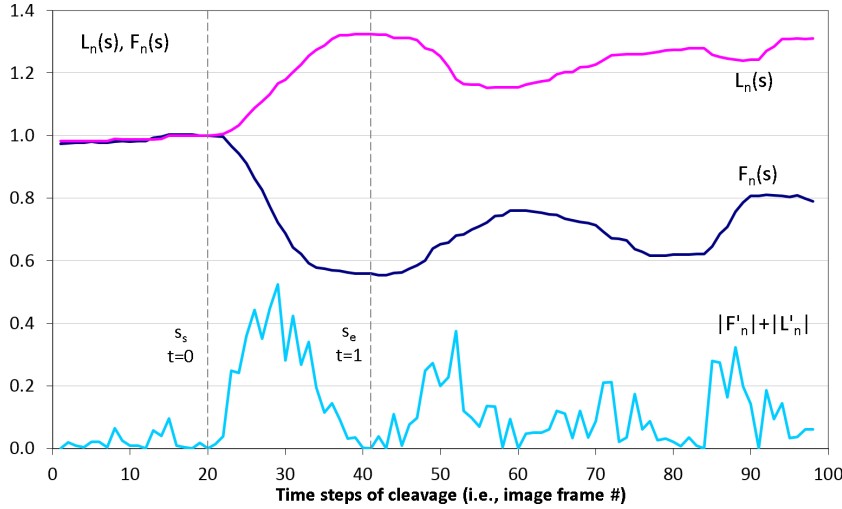

**Figure 3 Detection of the first embryonic division of *Sk* cells in image time series.** Median-smoothed time series of the furrow (dark blue) and polar (pink) lengths are combined to detect start $s_s$ and end $s_e$ time steps of the first embryonic cell cleavage using the sum of their derivative magnitudes $|F'(s)| + |L'(s)|$ (light blue). Based on $s_s$ and $s_e$, dimensionless time interval $t \in [0; 1]$ of the cleavage is introduced.

**Table 1 Example of conversion of physical dimensions to dimensionless units.**

| Scale | Cell cross-section | Furrow width | Spindle length |
|---|---|---|---|
| Physical space | 400 µm | 8 µm | 60 µm |
| Virtual model | 100 | 2 | 15 |

one-material, one-mechanism model, we iteratively refine and extend it by minimizing the deviation between computationally predicted and experimentally observed cell shape changes.

Cellular matter is approximated as an elastic (Hookean) material described by the piecewise linear stress–strain relationship (St. Venant-Kirchhoff material law) (*Ciarlet, 1988*):

$$\boldsymbol{\sigma}(\boldsymbol{\varepsilon}) = \frac{E}{1+\nu}\left(\boldsymbol{\varepsilon} + \frac{\nu}{1-2\nu}\text{tr}(\boldsymbol{\varepsilon})\mathbf{I}\right),\tag{1}$$

where $\boldsymbol{\sigma}$ denotes the Cauchy stress tensor, $\boldsymbol{\varepsilon}$ is the Green–Lagrange strain tensor and $(E, \nu)$ are the Young's modulus and the Poisson's ratio, respectively. The linear stress–strain is a basic property of mechanical continuum in the range of small relative deformations, i.e., $\boldsymbol{\varepsilon} \leq 0.05$. However, due to the fact that the strain tensor is a nonlinear function of displacement $\boldsymbol{\varepsilon}(\mathbf{u})$:

$$\boldsymbol{\varepsilon}(\mathbf{u}) = \frac{1}{2}\left(\nabla\mathbf{u}^{\text{T}} + \nabla\mathbf{u} + \nabla\mathbf{u}^{\text{T}}\nabla\mathbf{u}\right)\tag{2}$$

the equations of elasticity theory are, in general, nonlinear. In the range of small deformations, the higher order quadratic terms in (2) are usually omitted resulting a fully linear formulation of continuum mechanics with respect to the displacement.

The deformation of the entire cell is calculated numerically by integrating the partial differential equations of elastostatic equilibrium

$$\text{div}\,\boldsymbol{\sigma}(\boldsymbol{\varepsilon}) + \mathbf{f} = 0\tag{3}$$

under consideration of the boundary conditions and forces $\mathbf{f}$ that are given implicitly in the form of predefined boundary displacements, i.e., radial contraction of the equatorial cleavage furrow region. To solve the boundary value problem given by (1)–(3) for a 3D spatial domain, the Finite Element Method is applied in a way as previously described (*Gladilin et al., 2007*).

The above model of cell mechanics does not consider temperature as an explicit material parameter. We do not strive for formulation of a constitutive law with an explicit temperature dependency, because differently from classical engineering materials, temperature affects not only passive material properties of living cells, but also rates of all chemical reactions, which, in turn, may have an indirect impact on material cell behavior. Instead, we consider temperature as an implicit parameter which according to the literature (*Chan et al., 2014*) can be expected to influence canonic material parameters (i.e., cell stiffness).

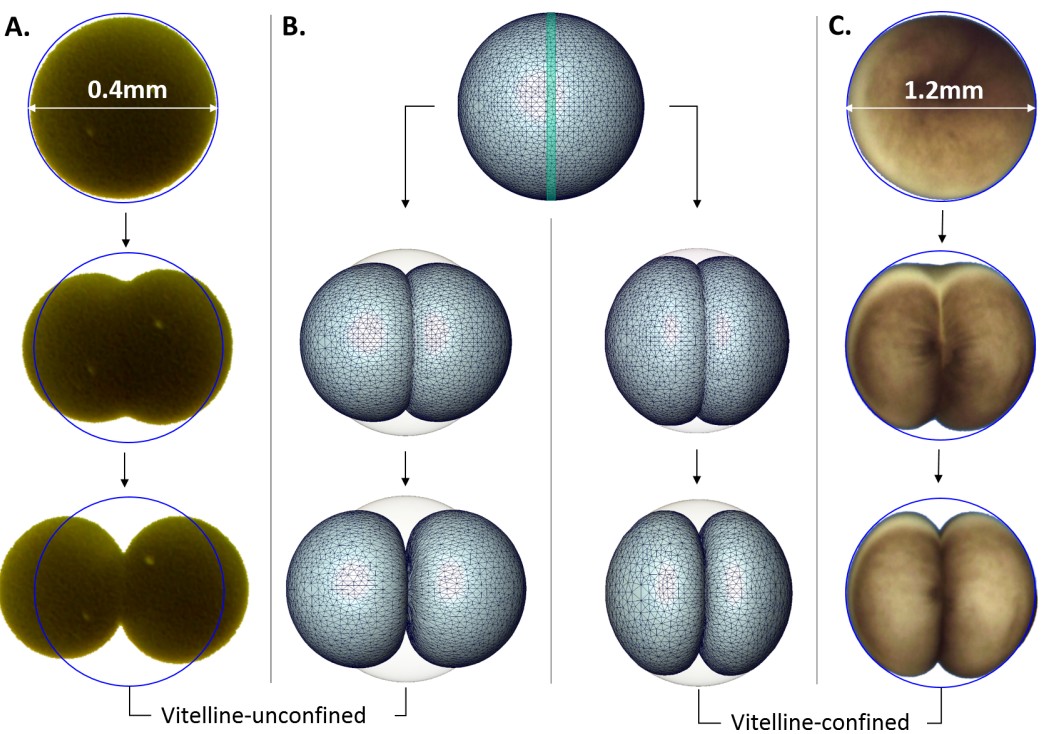

**Figure 4 Microscopic images vs. Finite Element simulation of the first embryonic cell division.** (A, C) show three sample stages of the cleavage of vitelline-unconfined *Sk* cell vs. vitelline-confined *Xl* cell (C) vs. Finite Element simulation (B) of Neumann (A) and sliding (C) boundary conditions given on the vitelline membrane. The green stripe in the equatorial plane of the initial FE cell model (B) indicates the cleavage furrow. Apparently, mechanical vitelline confinement essentially determines the cellular shape during the cleavage.

## EXPERIMENTAL RESULTS

### Effects of vitelline confinement on cytokinesis: plausibility simulation

Evidently, changes of the cellular shape during the embryonic cleavage are mechanically restricted by the vitelline membrane, see Fig. 4. Relatively loose vitelline confinement let dividing *Sk* cells freely expand along their polar axis (Fig. 4A), while *Xl* cells are bound to deform within a tight vitelline confinement during the entire cleavage (Fig. 4C). We wanted to understand how differences in boundary constraints affect mechanics of cell division. To this end, the boundary conditions corresponding to *Sk* and *Xl* type of vitelline confinement need to be appropriately incorporated into mechanical cell model. From the viewpoint of structural mechanics, boundary conditions on the outer surface of dividing *Sk* cells represent an example of the Neumann type of (free) boundary, while vitelline confinement of *Xl* cells can be described by the slippery (sliding) boundary condition which allows only tangential displacement along the boundary surface and forces its normal component to vanish, i.e., $\mathbf{u}\,\mathbf{n} = 0$. Figure 4B shows three characteristic exemplary steps in the Finite Element simulation of the cell cleavage for these two, considerably different types of boundary conditions, see the corresponding video sequences

(Videos S1 and S2) In both cases, an incompressible one-material linear elastic model of cellular matter is assumed and the boundary loads are given implicitly by successive contraction of the equatorial furrow region with the width indicated in Table 1. To minimize the error due to linear elastic approximation, the FE simulation of cell cleavage is performed in 10 successive steps, by each of which a relatively small rate of 5% furrow contraction is applied. For each next step of the FE simulation, the deformed cellular surface from the previous iteration is used to regenerate a consistent, high-quality tetrahedral mesh and to avoid undesirable artifacts due to largely deformed and deteriorated tetrahedrons.

In addition to striking differences between the geometry of vitelline-confined and vitelline-unconfined cell deformations, significant differences in the amount of mechanical energy required for these two types of cellular division can be expected. For this purpose, the volume integral (i.e., sum over tetrahedral elements) of the strain energy density is calculated (*Ogden, 1984*)

$$W = \int [\boldsymbol{\sigma} \, \boldsymbol{\varepsilon}] \, dV = \int \left[ \frac{\lambda}{2} (\mathrm{tr} \boldsymbol{\varepsilon})^2 + \mu \, \mathrm{tr}(\boldsymbol{\varepsilon}^2) \right] dV \qquad (4)$$

where $\lambda = \frac{E\nu}{(1+\nu)(1-2\nu)}$ and $\mu = \frac{E}{2(1+\nu)}$ are the Lame constants. For incompressible material (i.e., $\nu = 0.5$), the first term in (4) is neglected. Our numerical simulations show that cleavage of a vitelline-confined cell requires 4.8 times higher mechanical energy in comparison to the vitelline-unconfined case provided all other cell parameters (including cell size/volume) in both cases are equal.

As a result of tight vitelline confinement, the deformation of *Xl* cells is directed inwards and practically not visible in the outer cell contours. Therefore, all further experiments and model validations are performed using image data of *Sk* cells that are not constrained by vitelline envelope most time of the cleavage. By the end of cleavage, some elongated *Sk* cells can, however, come in contact with vitelline membrane, which restricts their further polar expansion. Due to correlation between vitelline cross-section and temperature, which will be discussed in more details in subsequent sections, the effects of vitelline as a mechanical obstacle are more frequent by low temperature conditions.

## Effects of furrow width and stiffness inhomogeneity: model sensitivity analysis

Geometrical and physical parameters of embryonic egg cells can considerably vary from species to species as well as during the cleavage time for the same egg cell. We want to estimate sensitivity of our computational predictions with respect to polar cell elongation in dependency on variable furrow width and cell stiffness that represent two key parameters of our contractile ring model of the cell cleavage. Previous works (*Schroeder, 1972*; *Zang et al., 1997*; *Foe & Von Dassow, 2008*) describe large changes of the furrow width (i.e., 3–17 µm) during the first embryonic cell division of different species. Remarkably, all authors make a similar observation that the furrow width reduces from the maximum value in the first half of the cleavage to the minimum value which is reached by the end

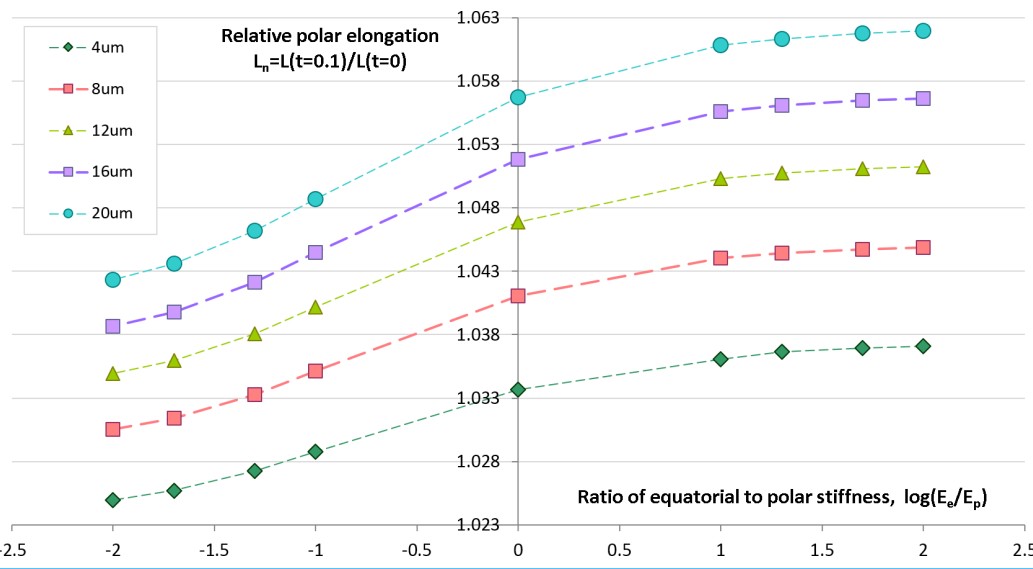

**Figure 5 Simulation of cell elongation for variable values of furrow width and polar stiffness gradient.** Dots show simulated relative cell elongation ($L_n$) corresponding to the first 10% contraction of the cleavage furrow as a function of the ratio between equatorial and polar cell stiffness ($\log(E_e/E_p)$) for four discrete values of the furrow width (4, 8, 16, 20 μm).

of cytokinesis. Inhomogeneity of stiffness between equatorial and polar regions has been contradictory discussed in the past. While *Matzke, Jacobson & Radmacher (2001)* reports increase of stiffness in equatorial furrow region, *Koyama et al. (2012)* makes completely opposite statements based on their image analysis and knock-down experiments. To analyze the effects of furrow width and stiffness inhomogeneity, we numerically simulate the cellular elongation resulting from the initial 10% contraction of the cleavage furrow. The computational simulation is performed for a range of reasonable values of the furrow width (4–16 μm) and stiffness gradient between equatorial ($E_e$) and polar ($E_p$) cell regions ($\log(E_e/E_p) \in [-2, 2]$). The result of these simulations are shown in Fig. 5 allow the following conclusions:

- The relative cell elongation triggered by the initial 10% contraction of the cleavage furrow varies between 2.5% and 6.2% within the entire range of reasonable values of furrow width and equatorial-to-polar stiffness gradient.
- Increase in furrow width results in increased polar cell elongation under otherwise same conditions.
- Increased stiffness of equatorial region (i.e., $E_e/E_p > 1$) leads to slightly larger polar elongation in comparison to the homogeneous case $E_e/E_p = 1$; softening of equatorial furrow region (i.e., $E_e/E_p < 1$) has the opposite effect.
- Twofold increase of furrow width has 10 times higher effect on polar cell elongation as twofold difference between equatorial and polar cell regions.
- Predictions of cellular elongation are, in general, not unique with respect to combination of model parameters.

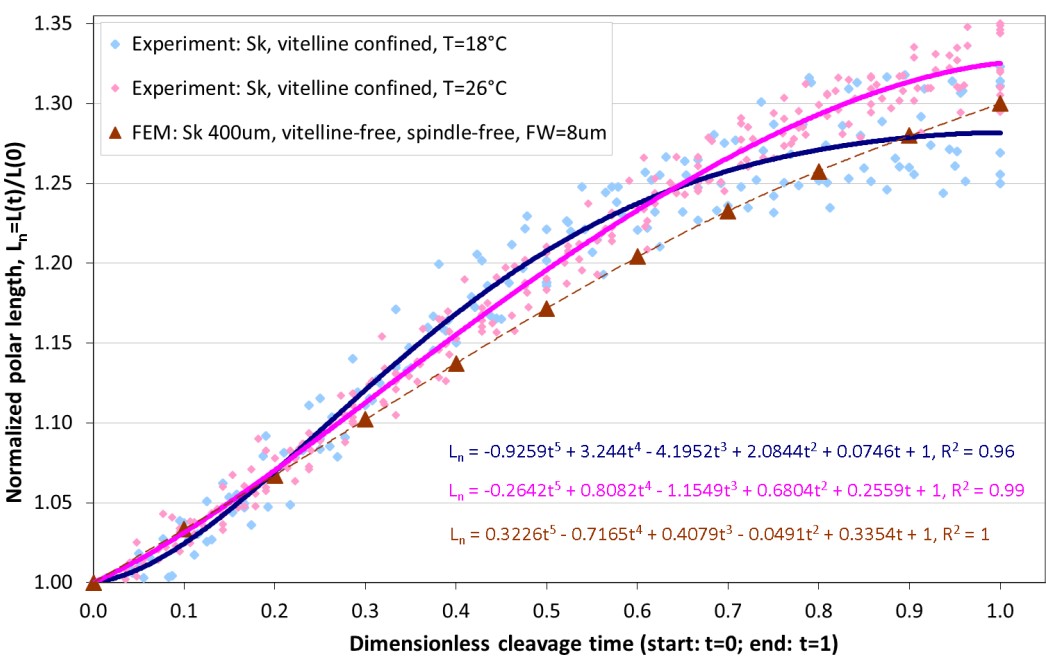

**Figure 6 Comparison of polar elongation measurements of *Sk* cells between two different temperatures (18 °C and 26 °C) vs. FE simulation of egg cleavage resulting from contraction of equatorial furrow.** Dots represent cumulative measurements of relative elongation of embryonic *Sk* cells under the given temperature condition as a function of dimensionless cleavage time $L_n(t)$. Each dot corresponds to simulated re. measured polar cell length $L_n(t)$ at a distinctive time point ($t = [0, 1]$) of the cleavage. Lines depict polynomial data fits.

## Effects of temperature on cytokinesis: experimental data vs. simulation

Altogether, from two *Sk* populations exposed to two different temperatures (18 °C and 26 °C), 10 and 16 cells are respectively selected for image analysis which is performed as described above, including segmentation of ROI, extraction of cellular contours and determination of the equatorial furrow $F(t)$ and embryonic polar $L(t)$ lengths for all time steps ($t \in [0, 1]$) corresponding to the first cleavage. For invariant representation of data, absolute lengths are substituted by dimensionless normalized values ($F_n(t) = F(t)/F(0) - 1$ and $L_n(t) = L(t)/L(0) - 1$) that have been used for quantitative description of cytokinesis in previous studies (*Hiramoto, 1958*; *Pujara & Lardner, 1979*; *Akkas, 1981*). Since dividing cells do not separate during the cleavage sufficiently enough to accurately detect their boundaries in the furrow region, the cross-section of equatorial contractile ring $F_n(t)$ is not precisely quantifiable for the entire duration of the cleavage. Consequently, the dimensionless cleavage time ($t \in [0, 1]$) is calculated from the time course of image frames as described in 'Image processing' and the polar embryonic length $L_n(t)$ is used for quantitative description of cell elongation. Figure 6 shows comparison of our measurements of polar elongation of dividing *Sk* eggs under two different temperature conditions (18 °C, 26 °C) vs. FE simulation of vitelline- and spindle-free contractile ring model of egg cleavage under assumption of the constant furrow width ($FW = 8$ μm or

2% of the initial cell cross-section). Comparison of experimental data and computational simulation of the *Sk* egg cleavage shows that:

- Cell elongations $L_n(t)$ for 18 °C and 26 °C do not exhibit statistically significant difference when compared as a whole (*p*-value: 0.88),
- At the end of the cleavage ($t \geq 0.7$), elongation of 18 °C *Sk* cells is significantly slower in comparison to 26 °C probe (*p*-value: 0.028),
- The result of our FE simulation matches well with experimental observations for both temperature conditions (*p*-values: 0.790 (18 °C) and 0.685 (26 °C), respectively),
- The linear pattern of simulated cell elongation differs from experimentally observed $L_n(t)$ curves that exhibit slightly nonlinear behavior.

## Effects of spindle elongation in small cells: extension of contractile ring model

The maximum elongation of *Sk* egg cells by the end of the cleavage amounts in average to 30.9%. Our computational simulation based on the contractile ring model with the unconstrained cell boundaries and constant furrow width predicts a similar elongation magnitude of 30%. However, significantly larger elongation rates of embryonic eggs have been reported for other species, i.e., 43.8% for *Cj* (*Hiramoto, 1958*), and previous theoretical models fail to explain such a large elongation of dividing cells on the basis of differences in passive material properties (*Pujara & Lardner, 1979*; *Akkas, 1981*; *He & Dembo, 1997*). As we have seen above, larger polar elongation can be theoretically attributed to larger width of the cleavage furrow. However, a strongly nonlinear pattern of *Cj* elongation reported in *Hiramoto (1958)* cannot be explained by larger constant or variable furrow width alone. Other sources of mechanical forces that are capable to drive such a large cell polarization has to be considered in addition to the contractile ring mechanism. Mitotic spindle elongation represents such an additional mechanism that is known to contribute to mechanics and shape changes of dividing embryonic cells (*Hiramoto, 1956*; *Schroeder, 1972*; *Wühr et al., 2009*). The reason why mitotic spindle can be expected to contribute to elongation of *Cj* to a significantly larger extent than in *Sk* cells, we see in different relative proportions between the cell size and the maximum spindle length which has been shown to have a principle upper length limit of 60 μm (*Wühr et al., 2008*; *Dumont & Mitchison, 2009*). Figure 7 demonstrates different relative proportions of the maximum spindle length to typical cross-sections of *Sk* (400 μm) and *Cj* (100 μm). Since small variations in spindle length and furrow width may have a large impact of elongation of small *Cj* eggs, they are both considered to be a variable parameters of our computational model that approximate ranges are estimated on the basis of available literature values (*Schroeder, 1972*; *Zang et al., 1997*; *Foe & Von Dassow, 2008*). Figure 8 shows the result of a stepwise multiparameteric fit of our FE model to the *Cj* elongation data (*Hiramoto, 1958*) in comparison to cumulative measurements and simulation of *Sk* embryonic cells. From the best fit between computational modeling and experimental *Cj* data, our simulation predicts successive decrease of the furrow width from 13 μm down

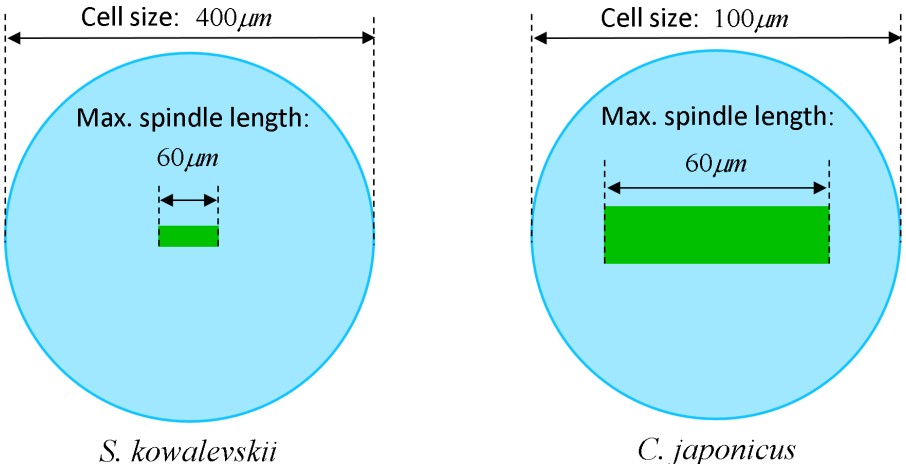

**Figure 7 Visualization of the relative geometrical proportion of the maximum spindle length (60 µm) to the size of *Sk* (400 µm) and *Cj* (100 µm) cells.** Effects of spindle polarization on mechanics of small *Cj* cells are significantly more pronounced in comparison to four time larger *Sk* cells.

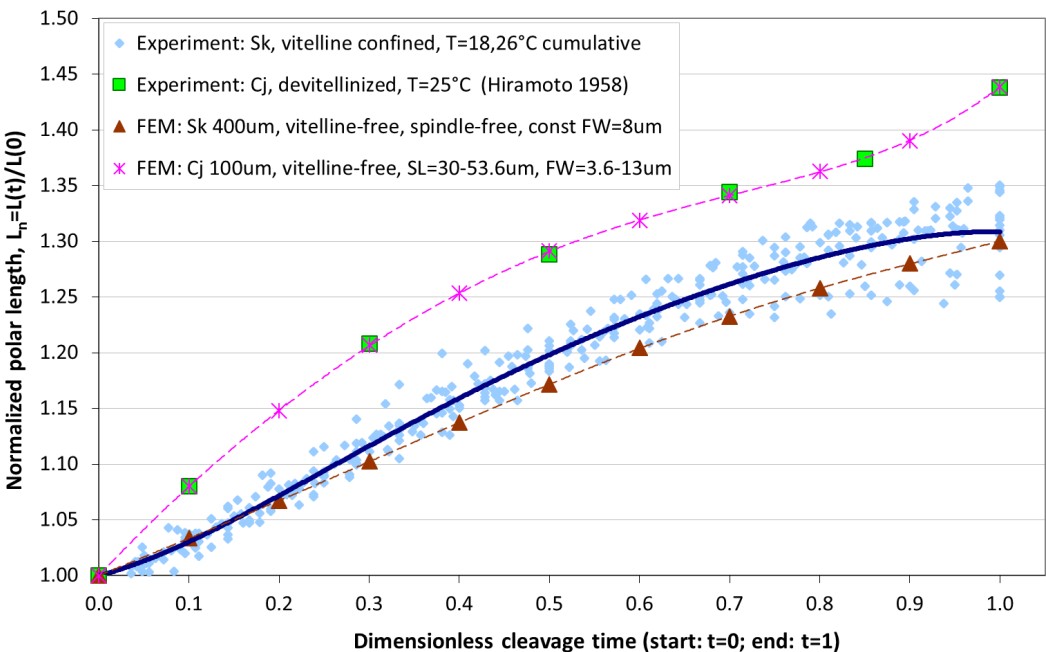

**Figure 8 Comparison of our cumulative measurements of polar elongation of *Sk* (400 µm) embryonic cells vs. *Cj* (100 µm) data (*Hiramoto, 1958*) vs. FE simulations of spindle-free (i.e., *Sk*) and spindle-extended (i.e., *Cj*) contractile ring models.** FE simulation of *Sk* cleavage is performed under assumption of vanishing effects of spindle polarization and constant furrow width ($FW = 8$ µm, i.e., 2% of the *Sk* egg cross-section), while division of the *Cj* cell is modeled by iterative fitting of variable furrow width ($FW = 3.6$–13 µm, i.e., 3.6–13% of the *Cj* egg cross-section) and spindle length ($SL = 30$–53.6 µm) to polar elongation measurements.

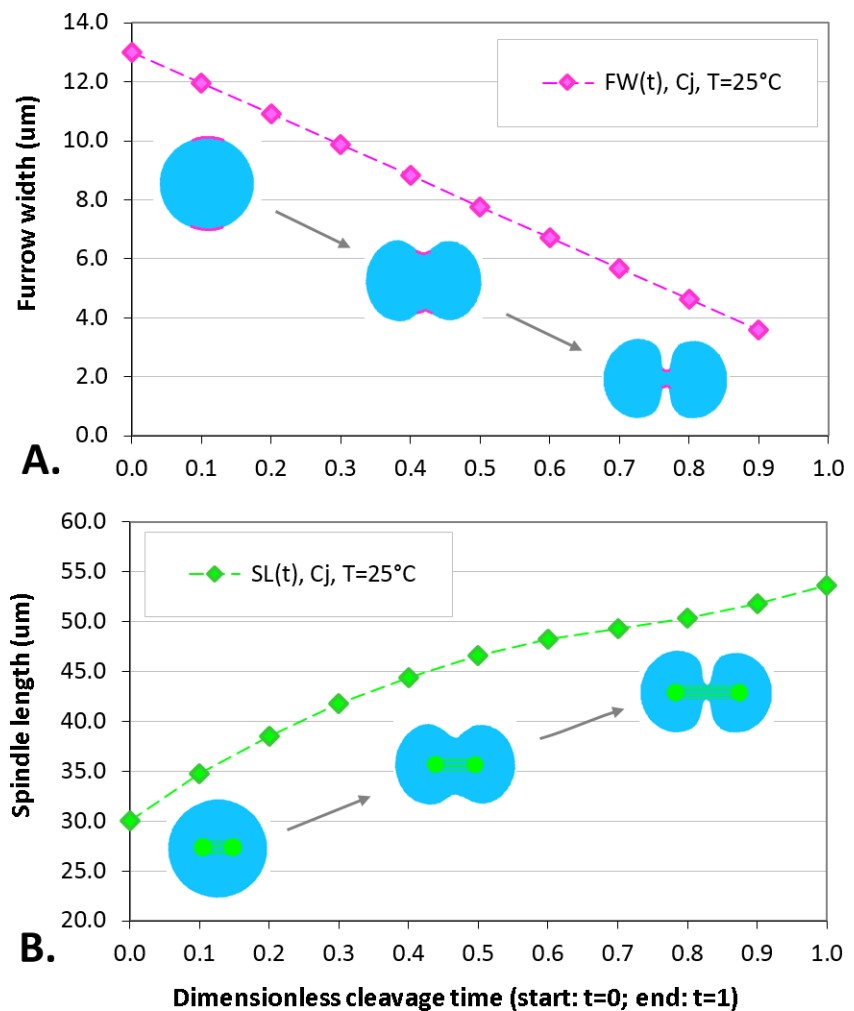

**Figure 9 Plots of furrow width (A) and spindle length (B) as a function of dimensionless cleavage time.** The values of FW and SL are estimated by a stepwise fitting of our FE cleavage model to *Cj* data (*Hiramoto, 1958*), cf. Table 2 (raws 5 and 6).

to 3.6 μm accomplished by slightly nonlinear elongation of mitotic spindle from 30 μm to 53.6 μm, see Fig. 9. Tabulated values of the relative egg elongation of all experimental and simulated data as well as fitting parameters as a function of dimensionless cleavage time are summarized in Tables 2 and 3, respectively.

## Effects of temperature on vitelline membrane

As we have seen above (Fig. 6), *Sk* cells exposed to lower temperature 18 °C experience an unexpected slow-down of the elongation rate at the end of the cleavage ($t = 0.8, 0.9, 1.0$) which is present neither in the experimental 26 °C probe nor in computational simulation. From the viewpoint of continuum mechanics, such an abrupt stop of polar cell expansion can be principally attributed to (i) a sudden decrease of the effective stretching force and/or (ii) mechanical obstacle which physically restricts free deformation of cellular matter. Both mechanisms can not be generally excluded. Contractile activity of actomyosin complex

**Table 2** Tabulated values of the relative polar cell elongation $L_n(t)$ as a function of dimensionless cleavage time ($t \in [0, 1]$): our measurements of $Sk$ (400 µm) vs. $Cj$ (100 µm) cells (*Hiramoto, 1958*) vs. FE simulations of spindle-free ($Sk$) and spindle-extended ($Cj$) contractile ring models, respectively.

| Cleavage time | 0.0 | 0.1 | 0.2 | 0.3 | 0.4 | 0.5 | 0.6 | 0.7 | 0.8 | 0.9 | 1.0 |
|---|---|---|---|---|---|---|---|---|---|---|---|
| 1. $Sk$, 18 °C | 1.000 | 1.024 | 1.070 | 1.121 | 1.168 | 1.208 | 1.237 | 1.258 | 1.271 | 1.279 | 1.282 |
| 2. $Sk$, 26 °C | 1.000 | 1.031 | 1.071 | 1.113 | 1.155 | 1.196 | 1.233 | 1.266 | 1.293 | 1.314 | 1.325 |
| 3. $Sk$, average | 1.000 | 1.030 | 1.072 | 1.116 | 1.159 | 1.198 | 1.232 | 1.262 | 1.285 | 1.302 | 1.309 |
| 4. FEM: spindle-free | 1.000 | 1.034 | 1.067 | 1.102 | 1.137 | 1.172 | 1.204 | 1.233 | 1.258 | 1.280 | 1.300 |
| 5. $Cj$, 25 °C | 1.000 | 1.081 | 1.149 | 1.206 | 1.253 | 1.290 | 1.319 | 1.342 | 1.363 | 1.391 | 1.438 |
| 6. FEM fit of 5. | 1.000 | 1.080 | 1.148 | 1.207 | 1.253 | 1.291 | 1.319 | 1.341 | 1.363 | 1.390 | 1.439 |

**Table 3** Tabulated values of furrow width and spindle length of a $Cj$ egg cell as a function of dimensionless cleavage time estimated by fitting of our FE model to polar elongation measurements (*Hiramoto, 1958*), cf. Table 2 (raws 5 and 6).

| Cleavage time | 0.0 | 0.1 | 0.2 | 0.3 | 0.4 | 0.5 | 0.6 | 0.7 | 0.8 | 0.9 | 1.0 |
|---|---|---|---|---|---|---|---|---|---|---|---|
| Furrow width (µm) | 13.0 | 12.0 | 10.9 | 9.9 | 8.8 | 7.8 | 6.7 | 5.7 | 4.6 | 3.6 | – |
| Spindle length (µm) | 30.0 | 34.8 | 38.5 | 41.8 | 44.4 | 46.6 | 48.2 | 49.3 | 50.4 | 51.8 | 53.6 |

has been observed in the second phase of cleavage not only in the equatorial furrow but also in polar regions (*Foe & Von Dassow, 2008*) which can effectively decelerate cell polarization. However, the fact that our $Sk$ cells are surrounded by the vitelline envelope let us firstly suspect the cause of abruptly reduced cell elongation in mechanical restriction of cell deformation by vitelline. Careful analysis of dynamic changes in vitelline membrane cross-section during the cleavage provides evidence for this assumption. Figure 10 shows time cources of the vitelline membrane cross-section to polar cell length ratio ($V/C$) of 18 °C and 26 °C $Sk$ cells. Surprisingly, the relative cross-section and dynamics of expansion of vitelline envelope during the cleavage strongly depends on temperature. As one may see, $Sk$ cells ($n = 23$) exposed to lower temperature (18 °C) exhibit significantly more narrow vitelline envelope (median($V/C$) = 1.32) already at the begin of the cleavage ($t = 0$) compared to population of cells ($n = 32$) observed by 26 °C (median($V/C$) = 1.48). In the late phase of cleavage ($t \geq 0.7$), the vitelline envelope of 18 °C incubated $Sk$ cells becomes a mechanical obstacle for polar egg elongation (median($V/C$) = 1.0). In contrast, 26 °C $Sk$ embryos do not experience such a boundary constraint (median($V/C$) > 1) and freely elongate up to the cleavage end. Comparison of the entire time course of $V/C$ ratios shows a statistically significant difference between measured populations of 18 °C and 26 °C incubated $Sk$ cells ($p$-value: 0.009).

## DISCUSSION

Despite an apparent geometrical simplicity, the first embryonic cell division represents a remarkably complex interplay of intracellular force generating, sensing and transmitting mechanisms with environmental conditions. A number of important factors, including mechanical vitelline confinement, environmental temperature and egg size relative to spin-

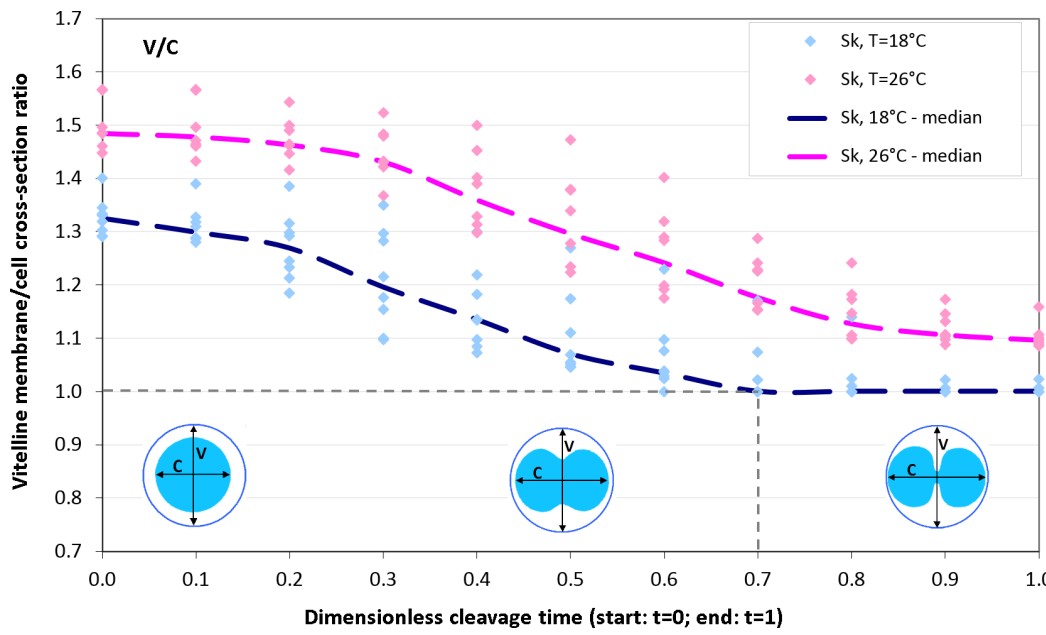

**Figure 10 Vitelline membrane cross-section to polar cell length ratio (V/C) as a function of dimensionless cleavage time ($t \in [0, 1]$).** In the late phase of cleavage ($t \geq 0.7$), polar elongation of 18 °C *Sk* cells is hindered by a tight vitelline membrane (i.e., median($V/C$) = 1.0). In contrast, 26 °C *Sk* cells do not experience such a boundary constraint (i.e., median($V/C$) > 1).

dle length has been considered in our study (see Table 4), some others have to be neglected for the sake of model simplicity. One of such major simplifications consists in assumption of homogeneous and isotropic cellular matter and neglecting of intracellular structures, such as yolk granules. *Sk* egg cells, at least the direct developing kind we consider here, are in fact 'somewhat telolecithal'—during cleavage stages, the vegetal hemisphere blastomeres are slightly larger than the animal blastomeres like in *Xl*. Nevertheless, we believe that our simplified symmetric model is capable to describe the essential properties of embryonic cells. This assumption is confirmed by our model sensitivity analysis which indicates higher impact of force generating/restricting components (i.e., cleavage furrow, mitotic spindle, vitelline) on cell mechanics in comparison to variation of material properties such as simulated axial gradient of cell stiffness.

In view of global nature of temperature effects on cellular metabolism and structure, differences in cytokinesis of *Sk* cells incubated at 18 °C and 26 °C appear to be surprisingly moderate. We interpret this result as an indication for existence of coordinated mechanisms of temperature adaptation that couple dynamics of cortical actomyosin machinery with mechanical properties of cytoskeletal fiber-rich cytoplasm in a way similar to recently suggested by *Turlier et al. (2014)*.

The results of our experimental and computational investigations provide evidence for a tight coupling of effects of vitelline confinement, environmental temperature and egg size on mechanics of the first embryonic cell division. We hypothesize that mechanical restriction of polar cell elongation by vitelline membrane contributes to proper regulation

**Table 4 Summary of experimental conditions (species, temperature) and measurements including egg cell size, maximum spindle length (60 μm) to cell cross-section ratio ($S/C$), maximum polar cell elongation ($L_n(t = 1)$), average vitellne membrane cross-section to polar cell length ratio ($V/C$) and type of vitelline confinement (i.e., loose ($V/C > 1.0$), tight ($V/C = 1.0$)) at the begin/end of the cleavage.**

| Species, T (°C) | Egg size | $S/C$ | $L_n(t = 1)$ | $V/C$ (begin/end) |
|---|---|---|---|---|
| *Xl*, 26 °C | 1200 μm | 0.05 | 1.0 | 1.0 (tight)/1.0 (tight) |
| *Sk*, 18 °C | 400 μm | 0.15 | 1.28 | 1.3 (loose)/1.0 (tight) |
| *Sk*, 26 °C | 400 μm | 0.15 | 1.33 | 1.4 (loose)/1.1 (loose) |
| *Cj*, 25 °C | 100 μm | 0.6 | 1.44 | None (devitellinized) |

of the embryonic cell division by activating mechano-signaling circuits and/or a direct mechanical feedback to spindle apparatus.

The role of intact vitelline confinement is apparently not limited to the first embryonic division and remains to be essential for subsequent development steps. *Sk* natural habitat is under shallow water in deposition of sediment carried by a river as the flow leaves its mouth. Thus, *Sk* gets exposed to natural temperature variation of 10 °C during one day. The embryo develops normally between 14 °C and 25 °C. Consequently, two temperature conditions investigated in our study (i.e., 18 °C and 26 °C) represent average and upper limit of the natural temperature range of *Sk*, respectively. While low temperatures (<14 °C) can be contra-productive for development due to global slow-down of kinetic rates, toxicity of moderately higher temperatures (>25 °C) is not self-explanatory. Based on our experimental results, rapid expansion of vitelline membrane and, as a result, lack of important mechanical clues can be assumed to play a role. In this respect, our results support previous observations that vitelline membrane provides an important micromechanical environment around the early embryo which absence or improper function impairs normal development (*Schierenberg & Junkersdorf, 1992*).

## CONCLUSIONS

In this study, we dealt with image- and model-based analysis of shape changes during cytokinesis of different embryonic egg cells that are exposed to different environmental conditions. In summary, the results of our experimental and computational investigations of the first embryonic cell division are as follows:

- Shape changes during the first embryonic division of relatively large eggs cells such as *Xl* (1,200 μm) and *Sk* (400 μm) are in good agreement with predictions of our computational contractile ring model under consideration of different boundary conditions due to tight or loose vitelline confinement.
- Our simulations predict higher energy demand for division of tightly vitelline-confined *Xl* in comparison to loosely vitelline-confined *Sk* egg cells. Thereby, our calculations account only for the mechanical deformation energy without consideration of higher friction of tight vitelline membrane in *Xl*. Consequently, real energy expenditure of tightly vitelline-confined cells is even higher.

- Sensitivity analysis of our computational FE model indicates that geometrical parameters of force generating structures (i.e., cleavage furrow width, spindle length) have stronger impact on elongation of egg cells in comparison to passive material properties (i.e., polar inhomogeneity of cell stiffness).

- Effects of environmental temperature on *Sk* cell elongation during the cleavage turn out to be rather moderate. However, temperature also affects the rate of vitelline membrane expansion. Here, we show that the low rate of vitelline membrane expansion at 18 °C leads to mechanical restriction and slow-down of *Sk* cell elongation in the late phase of the cleavage. In contrast, *Sk* cells exposed to higher temperature (26 °C) do not experience such mechanical constraint due to more rapid expansion of vitelline envelope.

- Large elongation of small egg cells such as *Cj* (100 μm) (*Hiramoto, 1958*) cannot be explained on the basis of the contractile ring mechanism only. We show that polarization of mitotic spindle, which maximum length is principally limited to 60 μm, can explain higher elongation rates of *Cj* in comparison to four times larger *Sk* cells. Furthermore, our model predicts progressive reduction of the furrow width during the cleavage of *Cj*.

Further quantitative investigations of embryonic egg cells of different species/sizes under different environmental conditions are required to generalize our findings that are based on exemplary analysis of *Xl*, *Sk* and *Cj* cell shape changes during cytokinesis. Furthermore, additional fluorescent imaging of all modeled subcellular structures would essentially help to furnish experimental proof of computationally predicted furrow width and spindle length changes in course of the embryonic cell division.

### Abbreviations

| | |
|---|---|
| *Sk* | *Saccoglossus kowalevskii* |
| *Xl* | *Xenopus laevis* |
| *Cj* | *Clypeaster japonicus* |

## ACKNOWLEDGEMENTS

Authors are grateful to Jessica Gray for generous help with acorn worm embryos, and thank Victor Luria for critically reading the manuscript.

### Funding

Leonid Peshkin was supported by the NIH grant R01 HD073104. The funders had no role in study design, data collection and analysis, decision to publish, or preparation of the manuscript.

### Grant Disclosures

The following grant information was disclosed by the authors:
NIH: R01 HD073104.

## Competing Interests

The authors declare there are no competing interests.

## Author Contributions

- Evgeny Gladilin conceived, designed and performed the computational experiments, analyzed the data, wrote the paper, prepared figures and/or tables, reviewed drafts of the paper.
- Roland Eils conceptualized the project, co-wrote the paper, reviewed drafts of the paper.
- Leonid Peshkin conceived, designed and executed the laboratory experiments, contributed reagents/materials/analysis tools, co-wrote the paper, prepared figures and/or tables, reviewed drafts of the paper.

## Data Availability

Raw data not included in the Supplemental Information (>2 GB) is available upon request to the authors.

## Supplemental Information

Supplemental information for this article can be found online at http://dx.doi.org/10.7717/peerj.1490#supplemental-information.

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
