# Peer review of "On the embryonic cell division beyond the contractile ring mechanism: experimental and computational investigation of effects of vitelline confinement, temperature and egg size"

_PeerJ, doi:10.7717/peerj.1490_

## Round 0.1 · original submission · Major Revisions

The paper is interesting, but according to the reviewers' comments, several important issues need to be addressed before the paper can be considered acceptable for publication. If you decide to resubmit an amended version, please answer all the comments written by the two reviewers.

Reviewer 1 ·

Basic reporting

The general organisation of the article, as well as the writing is well done. However, it would be advisable to show more information on what the figures show. For instance, in figure 3 it is not mentioned which eggs are being plotted (Saccoglossus or Xenopus). In this same figure, it would be interesting if they could show the measurements for both types of eggs, perhaps after normalising with respect to cleavage to compare both behaviours.
In figure 8, the color code is not explained.
The authors should also use scientific names when referring to the different species.

Experimental design

The authors begin by explaining the differences that vitelline membrane presence exerts on cell division between the two species (Saccoglosus or Xenopus), and they create models of how this physical barrier affects cell division. However, later on, they experience problems in distinguishing the furrow formation in the Xenopus samples due to the vitelline confinement. Is it not possible to segment the cells and predict where the furrow would be? This would allow to do more comparisons between the two samples.
Similarly, it is hard to compare to different species due to the intrinsic differences in embryonic development and composition in general. For instance, Xenopus eggs are telolecithal, so then the uneven distribution of yolk might already render completely different results as compared to isolecithal or centrolecithal eggs. Are Saccoglosus eggs also telolecithal? This should be at least discussed in the paper.

Validity of the findings

As mentioned above, in order to validate the effect of vitelline confinement in egg cleavage, it is important to include internal controls for both of the egg types. The authors should elaborate what happens to Saccoglosus eggs when they actually get in contact with the vitelline membrane. Does this affects cleavage timing, for example? For the Xenopus embryos, it would be interesting to see (if segmentation would allow it) how cleavage dynamics is affected by devitellinizing embryos.

Comments for the author

No comments.

Reviewer 2 ·

Basic reporting

See all comments in the General Comments for the Author section

Experimental design

See all comments in the General Comments for the Author section

Validity of the findings

See all comments in the General Comments for the Author section

Comments for the author

In this paper, Gladilin et al. investigate the role of temperature and vitelline membrane confinement in embryonic cell division. Using modeling and image processing tools previously developed by the authors, they investigate the first embryonic cell division in African Clawed Frog (X. laevis) and Acorn Worm (S. kowalevskii) embryos that differ in developing temperature and vitelline confinement properties.

First, they used a finite elements model with sliding and Neumann boundary conditions to represent the vitelline confinement properties of X. laevis and S. kowalevskii embryos. They found that the geometrical shape of cell division of their finite element model closely matched the experimentally observed shapes (Fig. 4). Then, they studied how temperature affects cleavage in S. kowalevskii embryos and compared experimental data with their model prediction.

The manuscript is presented in a straightforward manner. However, in my opinion, it has serious flaws that the authors must address before considering its publication:

1. The major problem of the manuscript is that the finite elements model does not take into account the experimental variables that they are considering, namely temperature and vitelline confinement (except for the comparison of boundary conditions in Fig. 4). Therefore, it is not clear how the finite elements model and the experimental data for different conditions can be compared because temperature and vitelline confinements are not tunable parameters in the model (see suggestion below). As presented, the comparison of experimental and modeling curves in Fig. 6 is not very useful.

2. One of the main claims of the paper is that temperature and the volume of the vitelline envelope are interlinked (Fig. 9). Unlike the authors (lines 258-263), I do not find this observation particularly surprising, because temperature likely affect density of the material in the vitelline envelope (making the embryo more confined at lower temperatures). The authors then interpret the observation that at 18C, the polar length plateaus at the end of the cleavage due to more tight vitelline confinement (Fig. 6, dark blue line). Although I agree that this is a possibility, the authors do not provide any evidence or rationale for this. There are two ways the authors could easily test this. First, they could use the model with hybrid boundary conditions, such that the model switches from free to sliding boundary conditions when the vitelline space is significantly reduced. In this situation, the model could also account for some of the non-linearities observed. Second, they could experimentally check if this plateau has anything to do with vitelline confinement by measuring the polar length during cleavage in devitellinized embryos at this temperature.

3. The article lacks biological insight/discussion into the possible advantage and/or disadvantage of vitelline confinement. Do the cell shape differences observed in vitelline-confined vs. vitelline-unconfined have any developmental consequences? What is the natural variation of the environment of these animals in the wild? Are they morphologically robust to these variations? These issues should be at least discussed briefly. Furthermore, temperature is known to affect the speed of development. In the manuscript, it is not possible to compare developmental speed at the different temperatures because time is scaled to the duration of the cleavage. It would be interesting to explore the issue of developmental time in these embryos.

4. In Fig. 8, the authors do not explain how they incorporated polar spindle expansion into their finite elements model. In any case, this mechanism to explain previously published experimental data from sea urchins embryos seems out of context and should be taken out unless it is explored further.

Other comments:

- In Fig. 4 and its discussion in the text, the authors claim that vitelline-confinement determine cellular topology. The term ‘topology’ is not appropriate as the topology in both cases is the same. This should be replaced by ‘geometry.’
- Most of the first paragraph in Section 3.3 should be moved to the methods section.

---

## Round 0.2 · Minor Revisions

As you can see, there are still several issues that need to be addressed before the manuscript can be deemed acceptable for PeerJ, yet both reviewers find the revised manuscript much improved. Please pay special attention to the points still raised by reviewer #1.

Reviewer 1 ·

Basic reporting

No comments.

Experimental design

No comments.

Validity of the findings

No comments.

Comments for the author

The authors have addressed this reviewer's concerns, by improving the description of their findings and the organization of the manuscript.

Reviewer 2 ·

Basic reporting

See General Comments to the Authors below

Experimental design

See General Comments to the Authors below

Validity of the findings

See General Comments to the Authors below

Comments for the author

The revised manuscript submitted by Gladilin et al. is a substantial improvement with respect to the original manuscript. However, I still do not feel that the paper shows a fair comparison between experimental and simulated data. Particularly:

- I still find unconvincing the comparison of the Finite Element model with the experimental data of the cleavage of S. kowalevskii (Sk) embryos at different temperatures shown in Fig. 6. Even if the authors decide not to explicitly include temperature in their mechanical model, I find it unsatisfactory that no effort is done to explain the different elongation behavior of Sk embryos at the end of the cleavage cycle at different temperatures. In Fig. 6, the authors compare a vitelline-free simulation with experimental data of vitelline-confined Sk embryos at 18˚C and 26˚C. I previously suggested to switch from free to sliding boundary conditions in the model after the polar length reaches the vitelline envelope in order to investigate if the observed slowing elongation at 18˚C could be explained by a tighter vitelline confinement at this temperature.

- In Fig. 8 the authors show that it is possible to fit a model with variable spindle length and furrow width to previously published experimental data of Cj embryos. The authors argue that no previous mechanical model have explained the observed large elongations of these embryos. Their fit of their model to the Cj experimental data is remarkably good, but this does not offer insights on what the key parameters underlying this elongation are. In my opinion, the paper should focus more on parameter exploration of the model rather than on model fitting. A detailed exploration of model parameters could show what are the roles of embryo size, spindle length and furrow width on the dynamics of cell elongation.

- In Fig. 10, the authors show that the vitelline membrane / cross section (V/C) ratio is slightly reduced at 18˚ C compared to 26˚ C. Is this reduction statistically significant? Although the V/C ratio is on average smaller at 18˚ C compared to 26˚ C, it is not clear if these small differences could account for the different elongation behaviors shown in Fig. 6. Note that in Fig. 6, slower elongation at the 18˚C embryos begins at time=0.7. It would be interesting to see if the V/C ratio in 18˚C embryos reaches its final value of 1 as soon as t=0.7, as one would predict if the slower elongation is due to vitelline confinement.
Alternatively, the slower elongation of Sk embryos at the end of the cleavage cycle at 18˚C may be due to other temperature-dependent metabolic effects that could affect cell mechanics. For example, perhaps the metabolic rates at 18˚ C cannot longer sustain the pace of elongation at the end of the cleavage cycle. Does the absolute cleavage cycle time at 18˚C much longer than at 26˚C?

In conclusion, although the authors present a much-improved version of the manuscript, a through exploration of the relevant model parameters is recommended.

---

## Round 0.3 · Minor Revisions

At this point, the paper is much improved, and in order to accept it for publication, we would like the authors to address the two points raised by reviewer #1.

Reviewer 1 ·

Basic reporting

No Comments

Experimental design

No Comments

Validity of the findings

No Comments

Comments for the author

No Comments

Reviewer 2 ·

Basic reporting

See comments below

Experimental design

See comments below

Validity of the findings

See comments below

Comments for the author

The authors have addressed most of the reviewers comments and the study deserves to be published.

My final minor comment is about trying to bring back the results of the study to the biology of these animals. The authors discuss a bit of this in the last paragraph of the Discussion section where they say: "We hypothesize that mechanical restriction of polar cell elongation by vitelline membrane might be of importance for normal embryo development." However, they do not speculate why this might be important. Particularly, if there are 10C variations in the natural environment of Sk worms and yet they develop normally, how is vitelline confinement important al lower temperatures?
In addition, it would be interesting to briefly discuss how adult size or proportions of Sk animals is affected by different temperatures. It is well documented that most animals grow to bigger sizes in colder environments (this is known as Bergmann's rule or temperature-size rule). Is there any difference in final animal size in Sk embryos raised at 18C and 26C during the embryonic stage? Tighter vitelline confinement at 18C will constrain embryo growth (at least along the polar axis) and we would intuitively expect smaller or disproportionate animals. This is the opposite of what is expected in the temperature-size rule.

---

## Round 0.4 · accepted · Accept

Your paper is now accepted for publication in PeerJ.

Reviewer 2 ·

Basic reporting

No comments

Experimental design

No comments

Validity of the findings

No comments

Comments for the author

The manuscript is acceptable for publication. The authors should check for small typos (e.g. in the last sentence of abstract it says 'challenging', it should say 'challenge').